# Business Model and Principles of a Values-Based Bank—Case Study of MagNet Hungarian Community Bank

**Zsuzsanna Győri** [1,*] , **Yahya Khan** [2] and **Krisztina Szegedi** [3,*]

1   Department of Management, Faculty of Finance and Accountancy, Budapest Business School,
    H-1055 Budapest, Hungary
2   Department of Business Economics, Faculty of Economics, University of Miskolc, H-3515 Miskolc, Hungary;
    yahyakhan89@gmail.com
3   Department of International Commerce and Logistics, Faculty of International Management and Business,
    Budapest Business School, H-1055 Budapest, Hungary
*   Correspondence: gyori.zsuzsanna@uni-bge.hu (Z.G.); szegedi.krisztina@uni-bge.hu (K.S.)

**Abstract:** The 2020–2021 global pandemic has brought significant changes to social and economic life. Companies must begin to rethink their business models and values to meet these new challenges. Given the process of intermediation, which has an indirect and catalytic impact, banks have a responsibility and opportunity to transform the economy by, for example, lending to projects that encourage decarbonization and/or green energy. The purpose of this paper is to examine the MagNet Hungarian Community Bank's approach as a values-based bank in order to compare how the operation of the bank differs from that of traditional ones—even if these apply the Corporate Social Responsibility approach. The findings of an exploratory study of MagNet's responsible and sustainable way of doing business can help other sectors and companies cope with the global crisis and be a part of the transition to an inclusive, fair, and decarbonised economy. The data for this study were collected using semi-structured interviews with eleven bankers and three customers of MagNet Bank in 2019, as well as bank documents. Using this information, we created a business model for the bank, using the Business Model Canvas method. Within the business model we highlighted how MagNet Bank integrates principles of the Global Alliance for Banking on Values, with the aim of truly integrating sustainability into the core of their corporate genetic makeup, instead of as a partial or insincere measure. With the projection of the models onto each other, we emphasize the role of values in the business model. The combination of the models indicates potential learning points for the further development and long-term success of the bank and serves as an example of good practices for others. This is especially relevant when considering the greater need for improved crisis and risk management due to the pandemic and for the integration of sustainability considerations into business operations which have increased the impetus in the financial sector towards sustainability.

**Keywords:** values-based banking; sustainable finance; financing decarbonization; Business Model Canvas; Hungary

## 1. Introduction

The development of responsible finance dates back almost thirty years, a significant result of which is the principle of responsible banking, which was established in 2019 by the United Nations Environment Programme Finance Initiative [1]. To achieve sustainable development goals (SDGs) and a sustainable economy, the European Commission defines the concept of sustainable finance as finance which makes sustainability considerations part of financial decision-making, with a focus on more climate neutral, low carbon, energy- and resource-efficient, and circular projects [2]. Since March 2021, financial market participants have had to disclose the impact of their financial decisions on sustainable development [3].

Since the 2008 financial crisis, attention has turned towards the social responsibility of the banking sector, and sustainable and ethical banks have taken a more prominent

role. Not only did the banking sector play a key role in the 2008 crisis due to toxic banking products, but recent decades have seen irresponsible behaviour on the part of traditional banks in many areas, examples of which include the London Interbank Offered Rate (LIBOR)-fixing scandal [4,5], Foreign Exchange Manipulation [6], corruption [7] and money laundering [8]. These activities have incurred significant penalties and reputational losses [9], stricter regulations, and even a reassessment of the social role of banks [8]. The sustainable financial sector requires banks that are responsible and ethical while being financially competitive as well as having a high degree of resilience to crises. The conventional banking system cannot cope with modern challenges, and thus, both a holistic approach and the integration of sustainability are needed for its stable operation [10]. Although the principles exist, two questions remain. First, how can banks put these into practice? Second, what specific business models can be successful?

Among the many definitions of Corporate Social Responsibility (CSR), ISO 26000 defines social responsibility as the "*responsibility of an organization for the impacts of its decisions and activities on society and the environment, through transparent and ethical behaviour that contributes to sustainable development, including health and the welfare of society; takes into account the expectations of stakeholders*" [11] (p. 3). There are few banks to date that have not faced the issue of social responsibility and have not integrated it into their activities to some extent. At the same time, there is criticism, as CSR does not lead to real change in a bank's business model.

Values-based banks promise a feasible alternative to traditional, purely profit-oriented, unsustainable banking. It is a growing international movement with an increasing number of institutions affiliating with the Global Alliance for Banking on Values (GABV). There are 65 banks worldwide that operate on the Triple Bottom Line Impact principle, treating the social and environmental benefits generated by their activities as equivalent to economic profit [12]. The mission of these banks is to "*support the transition to an inclusive, fair and decarbonised economy*" [13].

At the end of 2019, the Partnership for Carbon Accounting Financials was launched by bank associations such as the GABV. In 2020, they created the Global GHG Accounting and Reporting Standard, which outlines methods for measuring financed emissions of six asset classes: listed equity and corporate bonds, business loans and unlisted equity, project finance, commercial real estate, mortgages, and motor vehicle loans [14,15]. According to Peter Blom, CEO of Triodos Bank and Chair of the GABV "*financial institutions recognize that measuring financed emissions is a catalyst for action, regardless of their size, business model or where they are in the world. GHG* (greenhouse gas) *accounting provides crucial information to assess the resilience of portfolios to climate-related risks and identifies opportunities to finance the decarbonization that's so urgently needed for the transformation to a net zero emissions society*" [16].

Globally, GABV has more than 50 million customers, USD 200 billion in assets and more than 77,000 coworkers [13]. Even during the pandemic, they reached their 2020 plan to attract more members and expand their full membership by reaching approximately 70–100 million customers. This proves that a values-based approach can be successful during such a crisis, just as it was in 2008 [17]. As García-Muiña et al. [18] and Sági [19] note, the pandemic created an enormous need for assessing and communicating events and their impacts on organizations to their stakeholders. A potential way for detecting and assessing these impacts is to embed values in the business model, resulting in a new approach to business. Although all GABV members have the same mission, they have slightly different business models [20] in accordance with their different scopes. Exploring and examining these models can be of value to both theory and practice.

There are few scientific articles in the literature that examine the practice of values-based banks based on the declared values and principles [21] and even fewer that analyse it using a case study method. Such studies include that of Scheire and De Maertelaere [22], who investigate the business models of values-based banks by preparing case studies: their research is based on the 2007 and 2008 annual reports, and websites of the banks supplemented by a meeting of five member experts. Geobey and Weber [23] look at a credit

union's conduct and social impact disclosure practice, while Kaufer [24], Chew et al. [25], and Tan et al. [26] use qualitative interviews based on in-case analyses for a better understanding of a sustainable bank's motivational background [22–26]. The purpose of this study is to fill the gap in the literature and examine the common principles of values-based banks through the operation of the only values-based bank in Hungary, MagNet Hungarian Community Bank, as well as investigating its business model and the organizational culture that supports it. Our research questions are the following:

1.	How are the six principles of GABV incorporated/embodied in the MagNet Bank operations?
2.	How can the value creation process be described using the Business Model Canvas in the case of MagNet Bank?

In the first part of the article, we present the theoretical background of the topic, followed by the methodology used to answer our research questions. We then introduce the methodology and results of our study, followed by a discussion and conclusion. These results can contribute to the operational improvement of both the investigated bank and other values-based banks and provide a good practice example for embedding values into business models for more effective and efficient crisis and risk management.

## 2. Literature Review

### 2.1. Values-Based Banking—Meaning, Impacts and Relations to Performance

The financial market is the mediator between economic actors with surpluses and shortages. Financial intermediators transform money by matureness, size, geographic location, and risk. In this way, they have a major impact on the development of other economic actors and can catalyse and regulate the economic activities of others. This is also the reason why the banking sector, both nationally and globally, has a more consolidated regulatory framework (mostly in financial terms, rather than social or environmental ones) than other sectors. Nevertheless, the role of the sector in sustainable development is enormous [27]. Although the banking sector is not directly responsible for significant levels of pollution, it has a significant indirect influence on carbon emissions [28,29].

The 2008 crisis drew attention to ethical expectations toward banks, as it was due to their individual moral failures, ethical failures related to management or governance, and social ethics failures [30]. The attitude towards *"gambling with other people's money"* [31] (p. 372) contributes to the uncertainty and the low level of trust in financial institutions [32]. There have been many attempts to rebuild confidence in the banking sector and prevent similar crises from occurring through stricter regulations to minimize risk, compliance functions within the banks, the emergence of ethical fair conduct and transparency principles, and measures to improve financial culture [33,34]. According to a recent study, despite efforts, there has been a regression in regulation, and much of the financial sector is still operating in an irresponsible and unsustainable way, which could cause another crisis [35]. In order to avoid this, some changes in economic logic and practice are needed. For example, one potential and recommended change is to build environmental risks into the cost–benefit analysis. An example of this would be to price and embed high environmental risks, related to carbon emission and climate change, into financing as a cost [36]. When considering changes to business models, the global spread of COVID-19 has had a significant impact on business actors such that *"the pandemic has distorted the organizations' business models and has substantially changed the value* creation processes" [18] (p. 1).

The concept of CSR has become a focus both for professional and academic scholars in business management, alongside the belief that CSR initiatives are contributing not only to the sustainability of businesses but to society as a whole. In 2009, the United Nations (UN) established the Sustainable Stock Exchange Initiative (SSE) with the goal of creating sustainable global finance while considering economic, social, and environmental issues, focusing on sustainable investments and business strategies [37]. The response of the banking sector to CSR challenges regarding environmental and social issues came relatively late when compared to other sectors [38]. However, Scholtens [39] argues that

banks are more engaged in financing activities and sustainable development, as society demands they be socially responsible and transparent. Many banking activities can affect society, either directly or indirectly, such as credit granting, risk management, assets management, and the operation of payment systems. Thus, the application of CSR as a strategy is crucial in order to improve banks' relationships with their stakeholders and enhance their performance and reputations [40]. The advancement of CSR has been greatly facilitated by the findings of several recent studies which have found that, in part or in full, a positive relationship exists between CSR and the financial performance of the banking sector [41–43].

Different terms are used to describe those banks that integrate ethical considerations into their operations: social banks, responsible banks, ethical banks, green banks, alternative banks, sustainability-focused banks, values-based banks [24,44–48]. In contrast to conventional banking, which only focuses on profit maximization, social banking or alternative banking conducts business operations in a way which constructs social, environmental, and sustainable benefits for society. This is performed through products and services such as the provision of loans and finances to green projects (renewable energy schemes), social enterprises, NGOs, and social housing projects [49]. Similarly, green banking is an environmentally friendly banking initiative, motivating clients to reduce their carbon footprints through business operations. Green banks function in the same way as other banks while considering environmental implications and encouraging decarbonization and the conservation of natural resources [50]. Hernaus [51] identifies some key differences between values-based banks and traditional banks. The development driver of alternative banks, which are mostly local banks, is not profit but ESG trends. Traditional banks may be varied (international, national, local) and offer modern and risky services (e.g., derivatives), while alternative banks simply offer classical banking services such as deposits and loans. Urban and Wójcik note that, despite some examples, ethical banking is still a niche market [52]. A further problem is that, despite the confusing variety of terms and standards associated with sustainable finance [53], sustainability initiatives in the banking sector are still peripheral and there is no real integration of sustainability principles into mainstream banking activity [52].

Values-based banking works sustainability into the core operations of the bank, and this for more than just typical CSR topics such as donations to charity, paper reduction, or equal opportunities for the workforce. Since the core business of a bank is financial intermediation, the sustainability context should define this as well. Banks would provide loans for creating a social, environmental, or sustainable benefit, and collect deposits where they guarantee the sustainable, socially appropriate utilization of money and serve real human needs by providing account services for NGOs or impoverished people.

The history of the first ethical bank dates to the late 19th century, but most such institutions were established in the 1980s and 1990s [54]. According to the GABV, " *members focus on a shared mission—to use finance to deliver sustainable economic, social and environmental development, and to help individuals fulfil their potential and build stronger communities*" [47] (p. 3), this is manifested in the six principles of GABV (Figure 1). The Triple Bottom Line approach is just one of the six principles that its banks hold. It is also important that they serve the real economy, establish a long-term relationship with customers, are resilient to long-term crises, have transparent and inclusive governance, and incorporate these principles into the bank's organizational culture [55].

Although CSR is commonly associated with increased costs, resulting in decreased competitive advantage, research results have in fact shown the opposite to be true: responsible companies perform better in financial terms because their main stakeholders are more loyal and trusting.

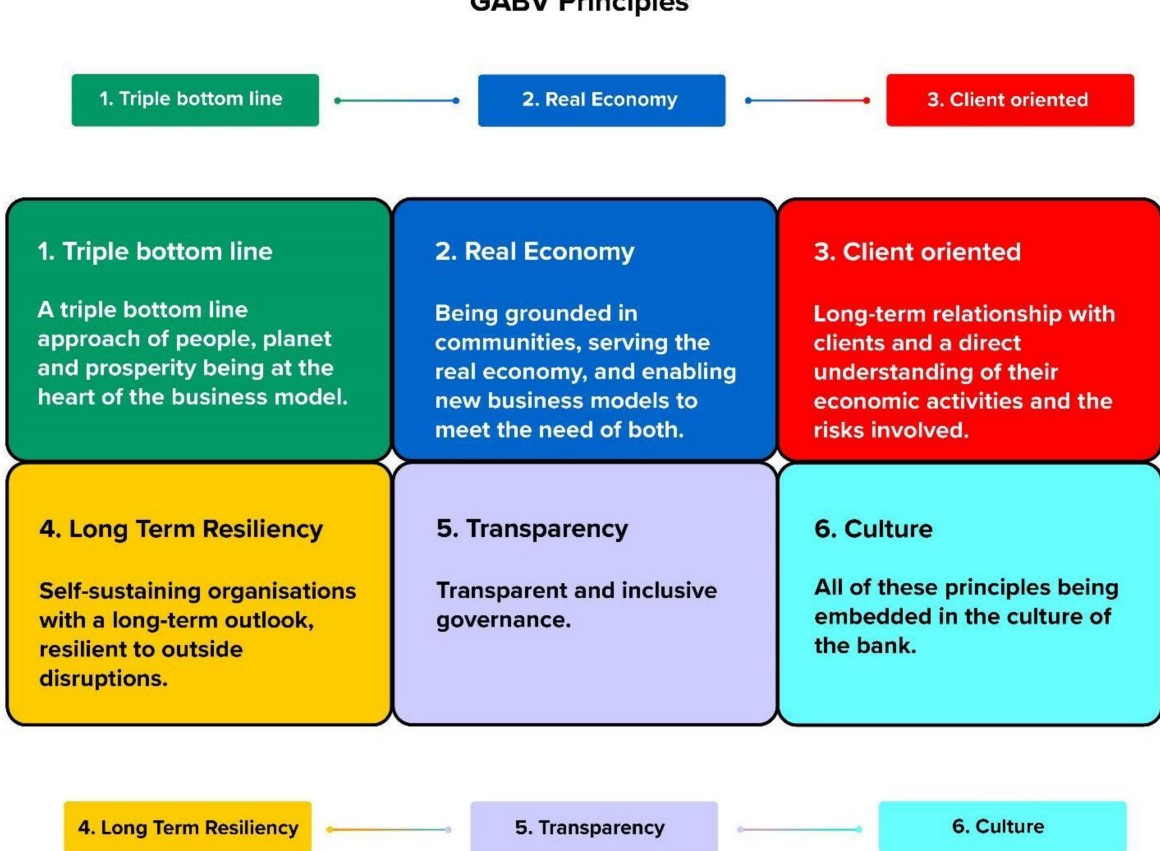

**Figure 1.** Principles of Values-Based Banking. Source: Authors' figure based on http://www.gabv.org/about-us/our-principles (accessed on 17 January 2020).

When it comes to clients, banks have two major sources of growth: increasing the base of conscious customers looking for conscious financial service providers, and clients excluded by other banks (NGOs, impoverished and disadvantaged people, or communities). Members of GABV had much higher growth in loans, deposits, assets, equity, and total income compared to those of traditional banks from 2006 to 2015, especially since the crisis began (growth of total income is 0.5% for traditional and 7.6% for values-based banks between 2011 and 2015) [56] (p. 8).

Authorities impose fewer fines due to legal and compliance issues to values-based banks compared to traditional profit-seeking banks. This is thanks to lower risk-taking, for example, lower levels of complex and deceptive products, such as derivatives and credit cards, within their assets [56]. Overall, this results in less volatility (0.26% compared to 0.35%) and, surprisingly, a higher level of returns on assets (0.65% compared to 0.53% from 2006–2015 [56]. GABV found values-based banks operate in numerous markets, serve diverse needs, and use a distinct business model, claiming that "*they are growing because they provide economically viable banking alternatives focused on the needs of society thereby creating a more diverse financial ecosystem.*" [56] (p. 3). The GABV found that, compared to the largest banks in the world they have the same strategic bases, but are more resistant to crises, and deliver better and steadier financial results.

Halamka and Teplý [57], in their quantitative study, compare the financial performance of 69 values-based banks with conventional banks. They find that values-based banks tend to have a higher percentage of net interest income as a share of total revenue, that their return on equity (ROE) is higher and that, despite their higher risk profile, ethical banks maintain lower and more stable figures for loan losses. They did not find a significant correlation with other performance indicators, which means that values-based banks, though they do not perform better, at least do not perform worse than mainstream banks.

A recent study comparing the economic performance of traditional financial institutions and values-based banks operating in Europe finds that GABV banks have performed financially better in the past 10 years than traditional banks. The study concludes that while many large traditional banks had to be rescued after the crisis, no values-based banks had to be rescued. It also finds that between 2007 and 2017, European values-based banks have recorded returns three times higher than those of mainstream banks, with an average annual profitability (defined in ROE) of 3.98% against 1.23%. They also note that assets, deposits, loans, and equity of ethical banks increased by 9.66% annually, compared to an annual performance of −1% on the part of mainstream banks, and that values-based banks also performed better against Environmental, Social, and Corporate Governance (ESG) criteria, both in material and immaterial terms, including factors such as access and affordability, labour practices, diversity and inclusion, or data security [35]. Table 1 summarizes the main differences between traditional and value-based banks.

**Table 1.** Major differences between traditional and values-based banks. Source: Authors' own table based on [21–31].

| Factor | Traditional Banks | Values-Based Banks |
|---|---|---|
| Approach | Shareholder | Stakeholder |
| Focus | Achieving Economic value (profit) | Repurposing finance based on SDGs, creating environmental, social and economic value |
| Basis of value-creation, criteria of banking practices' and results' evaluation | Annual revenue and profitability, shareholder value | ESG criteria for contributing to sustainable development |
| Geographical location/focus | International, national, local | Local |
| Products and services | Modern, risky | Classical, risk-averse |
| Responsibility | Only economic and legal responsibility and/or CSR outside the core business | Integrated into the core business |
| Role | Financial intermediation | Sustainability intermediation |
| Debtors and Creditors | Mainly as individual clients | Active decision-makers and actors of the community |
| Crisis resilience | Non-resilient | Resilient |

*2.2. Business Model Canvas*

Business modelling is a popular method for business planning and can also help delve into the essence of certain ventures. Of the many modelling tools, we have chosen to work with the Business Model Canvas (BMC) [58]. Though this model has its pitfalls, it also has undeniable advantages [59] as "*it is designed to convey the essentials of an enterprise, quickly, simply, and in a visual format*" [58] (p. 5). Contrary to a business plan, which is a long document with a lot of data, the BMC allows us to see all of the information deemed significant on a single page. In formalizing a business model using the BMC, we are able to observe the main parts of a business and their interdependence, which gives us an insight into the company's business framework and structure. It is also a flexible tool for planning where one or more building blocks are easily modified as needed.

The BMC can also be used to examine values-based banks [22]. We will apply the tool for the presentation of the main points of MagNet Bank's conception, to articulate the activities and business structure, and most importantly, the value proposition of values-based banks. Some studies attempt to create business models that integrate sustainability considerations and create a specific Ecocanvas [60], circular business model [61], or sustainability business model [62], extending value creation to stakeholders beyond customers [63]. One of the latest, complex approaches including the economic, social, and environmental dimensions of sustainability [18,64] is the Triple Layered Business Model Canvas (TLBMC), which interprets the original BMC as the economic dimension of sustainability, adding two additional layers to it, the environmental layer, representing the life-cycle approach, and

the social layer, representing the stakeholder approach [18,64]. The practical application of this model has already been tested in many sectors [18,65–71]. Although we consider the new variations of the tool to be promising for the future, our goal is to specifically use the original BMC [58] which is a widespread and well-known business modelling tool that can stress the importance of sustainability issues in a transparent, distinct way even for mainstream economists, bankers and decision-makers.

The nine building blocks of the BMC show the logic of how a company intends to operate (Figure 2).

## Business Model Canvas

**Figure 2.** Business Model Canvas. Source: Author's own figure based on [58]. Adapted with permission from ref. [58]. Copyright Year: 2010 Copyright Owner's Name: Osterwalder, A.; Pigneur, Y.

At the centre of the model, we find 'Value Proposition', which is the offer the firm makes and " *the reason why customers turn to one company over another*" [58] (p. 22).

The 'Customer Segments' section represents all the clients who gain value from the company (including those who do not pay for a certain product or service). 'Channels' and 'Customer Relationships' connect them with 'Value Proposition': 'Channels' describes the physical points where the business interacts with its customers, while the 'Customer Relationships' section outlines the type of relationship, from personal assistance, through automated services, to cocreation of value. The 'Revenue Streams' shows the transaction and recurring revenues from different customer segments.

The left side of the figure includes 'Key Resources', which represent the most important assets, and 'Key Activities', which represent important processes of the business. These are absolutely indispensable for value creation. Since the value creation process does not start with the specific enterprise (it has suppliers in the value chain), and it needs other external resources and services, the BMC also identifies the 'Key Partners' in the value creation process. While the whole canvas shows the structure of the business, the left side presents

the infrastructure of it and ends with the 'Cost Structure' that describes all costs incurred to operate the business model.

## 3. Materials and Methods

*Research Questions and Case Study Method*

We aim to understand the value creation of MagNet Bank, a member of GABV, and the only values-based bank in Hungary. They have been operating profitably since 1995 and are the largest financial institution with 100% Hungarian private ownership (since 2013). The company size (based on an annual report data from 2018) can be characterised as follows: balance sheet size: HUF 151 200 million (appr. EUR 469 million); share capital: HUF 8 070 million (appr. EUR 25 million); number of employees: 207.

MagNet has seven equity partners, and the functions of activity (business intelligence, treasury, sales, customer relations, operations, IT and development, legal issues and compliance, finance, HR, marketing, and community development) are divided among them. This means that the partners are actively engaged in the bank's operations and can enable the realization of values-based principles. "*Among owners/equity partners/board of directors a special position is set up for the Community Development director that overviews HR, marketing, Civic and community relations areas primarily and intervenes in all other areas if relevant. This function and position secure constant reflection of the board on values-based banking principles and their application in all actions*" [72].

In 2018 MagNet won the Socially Responsible Bank of the Year Award for the seventh time in the MasterCard Bank of the Year competition. Their slogan, "*My money builds*", expresses their desire to create a healthier connection between their clients and partners and money and its use for the sustainable development of the real economy. Their path can serve as an example and provide many learning points for other values-based banks or for responsible companies in other sectors. Traditional banks may also learn from them as the challenges of sustainability or the present global pandemic become more and more significant and urgent. Our research questions are the following:

1. How are the six principles of GABV incorporated/embodied in the MagNet Bank operations?
2. How can the value creation process be described using the Business Model Canvas in the case of MagNet Bank?

Yin defines the case study research method as "*an empirical inquiry that investigates a contemporary phenomenon within its real-life context; when the boundaries between phenomenon and context are not clearly evident; and in which multiple sources of evidence are used*" [73] (p. 23). He suggests that a general view on a special topic can be gained by investigating a special example. "*It can be considered a robust research method particularly when a holistic, in-depth investigation is required*" [74] (p. 1), as the method helps examine the topic from different viewpoints [75].

Tan, Chew and Hamid [26] use the case study method in order to obtain a better understanding of a sustainable banking operation's motivational background based on 35 qualitative interviews. This specific study does not examine the business model; however, as mentioned in the introductory paragraph, such studies have been conducted. Using a combination of the two cited methods, we examine the business model of one specific bank, but in a more in-depth and comprehensive way, by including the cost and revenue parts of BMC and using more primary data. Our analysis highlights the similarities of MagNet to the two cases in Scheire and De Maertelaere's work [22]: Triodos Bank and Banca Etica. Both are values-based banks that are much larger than MagNet Bank, with one being present in five countries and the other in two. At the same time, we examine the bank through a different framework, that of the six principles of GABV. We then connect the two theories to obtain a more comprehensive understanding of the motivations, activities, and value creation of MagNet Bank.

Our case study is based on primary data collected through qualitative interviews with 11 bankers (including equity partners) from different departments and three customers with

connections to the bank. The semi-structured interview includes open-ended questions to capture the interviewees' opinions regarding the research questions. We include the interview questions in Appendix A. The interviews were conducted in person before the pandemic period from July to August 2019. The sample includes managers and branch managers of the bank's functional areas, as well as representatives of the main customer categories (small business, NGO, and individual). The participants' consent was obtained prior to the conduction of the interviews. The interviews were transcribed, summarised, and categorised into themes based on the key elements presented in BMC [58]. Secondary data such as academic books and journals, website publications, MagNet Bank's annual reports and website were reviewed to supplement information obtained from the interviews.

## 4. Results

### 4.1. The Business Model of MagNet Hungarian Community Bank

"*A business model describes the rationale of how an organization creates, delivers, and captures value*" [58] (p. 14). Our study pays particular attention to the value capturing aspect of this definition, as values-based banks define value in a way which is different from traditional banks. "*It is banks embracing a viable model that strategically takes a longer-term view of profit and prosperity. For sustainability-focused banks, profit is a result of sustaining and growing value in the real economy and healthy communities, not an end goal*" [56] (p. 3). MagNet defines value as being "*at the heart of its business model and being created at all levels of our organization, through the products and services offered, and through the involvement of employees, customers, social entrepreneurs, and NGO partners. With our financial services, we work day-in day-out towards the establishment of a transparent, value-centric society*" [72].

#### 4.1.1. Customer Segments—Green, Decarbonization Project Owners, and Customer Assessment Based on MagNet Impact Scorecard

There are two main groups of customers: borrowers, on the asset side of the bank's balance sheet, and depositors, on the liabilities side. Individuals, companies (especially SMEs), and as a specialty, civil society organizations, can have access to financing from the bank. On the depositor side, offerings range from traditional deposit and savings accounts, to everyday payment services, to internet and mobile banking. The real specialty is to connect the two main groups through values-based intermediation.

The bank serves individuals, entrepreneurs, projects, and organizations that face difficulties in obtaining loans through the traditional banking system. An SME entrepreneurial client commented: "*previously I had not heard of MagNet Bank. I wanted to take out a loan, but I was not able to do so anywhere since possibilities decreased due to the crisis. MagNet Bank, however, offered me a loan to start my business.*" The bank adopts very different lending criteria compared to mainstream banks, wherein they decide to finance undercapitalised civil society organizations and social enterprises, or innovative ecological projects, such as green and low-carbon energy systems, which bear a high risk [76]. One civilian client claims that when they took out their loan, "*bank financing of foundations and NGOs was virtually non-existent*", but that "*since then, they [MagNet Bank] are practically the largest civil society financiers in Hungary.*"

Besides the classic indicators, the level of social and environmental utility is considered very important among the credit scoring factors of MagNet Bank. In the MagNet Impact Scorecard, which is MagNet Bank's self-developed, self-reported audit tool for quantifying the sustainability impact of an organization and/or a project [77] the governance, the institutional, the economic, the social, and the environmental impact of the specific company and the given project are assessed, including their carbon emission and initiatives to reduce it. Recently, the Hungarian National Bank issued a so-called green recommendation, which focuses on green loans issued by banks. MagNet is at the forefront of this initiative. In recent years, they have been able to grow in the solar sector. Their exposure to the green energy sector is HUF 10 billion, which represents more than 16% of their HUF 60 billion

corporate loan portfolio. A larger bank may not be able to achieve this high ratio, but MagNet's smaller size and values-based approach make it a possibility. This is an example of how the bank has a larger than average positive impact on decarbonization with the assessment and selection of creditors.

These features are the reason why the bank attracts a lot of customers looking for useful and sustainable financial services—similar to Triodos and Banca Etica [22], where the money of well-informed and highly educated depositors also flows to borrowers in specific sectors such as bioagriculture, healthcare and social services, environmental protection and nature conservation, culture and education, research and development, green energy, and job creation in the case of MagNet Bank [78].

### 4.1.2. Value Proposition

A key value is prudent banking, as a customer commented: "*this is a bank, so I expect it to be reliable, and my finances to be in order. The things MagNet Bank offers beyond that are extras. They organize events that I attend because I'm interested in the topics, I'm open to new things.*" According to one of the managers interviewed: "*in terms of its basic operation MagNet Bank is just like any other bank.*" Another manager also noted: "*I believe the most important thing about a bank is for it to operate like a bank.*" A branch manager stressed: "*bank products should work the same way as in any financial institution. Of course, MNB* [viz. Hungarian National Bank] *regulates all financial institutions the same way, within a fairly strict framework.*"

The bank combines traditional and non-traditional banking activities. There is an economic reason for this, as one manager put it, the bank "*can't sustain [itself] with community products alone.*" At the same time, with its special products and services, MagNet is not a standard bank: "*Our main vision is to maintain the uniqueness that MagNet Bank represents in the current banking market while providing potential.*"

Value creation is based on TBL and the opportunity of being in a community of environmentally and socially engaged people. The bank takes responsibility for the education of consumers on social and environmental issues. One manager describes the mission of MagNet Bank as follows: "*MagNet Bank develops products that make its customers responsible. This means developing financial awareness and contributing to a democratic, sustainable, harmonious society through widespread social awareness.*" Another manager mentions the term sensitization: "*With our community products ... we try to ... sensitize our customers, that is in our social impact, and social assistance ... we strive to take on a role that will make this society or this Earth a better place.*"

In 2019, under the Climate Change Commitment Initiative, MagNet committed to assessing and disclosing the climate impact of their portfolio of loans and investments within a period of three years, and ultimately, to ensure the climate impact of their loans and investments are in line with the Paris Agreement [79]. The first step is the above-mentioned engagement towards green energy projects within the corporate loan portfolio.

Furthermore, MagNet keeps the focus of its activity on its intermediation function and has direct investments in the real economy, in real society and not in the financial markets, apart from liquidity placements with other banks or investments in government bonds. The bank avoids repackaged, complex, and unclear, or unethical financial instruments such as derivatives or credit cards, which partly explains their crisis-resistance. As one of the interviewed managers claims: "*If such an insurance product were added to the products and our branch sold it with a lot of bonus pressure, I think it would be perceived very negatively ... every second customer would say that it stuck out as a sore thumb, and that we were just the same as any other bank.*"

### 4.1.3. Customer Relationships

Customer Relationships are defined by transparent information and active involvement of depositors in lending and activities that support the community. With a values-based intermediation and community donation program [80], the bank enhances awareness amongst clients about the impact of their financial decisions and builds a community with

them. This is similar to Triodos, where a Charity Saving Account exists and transparency is assured via the information available on the company's websites [22]. This community develops together, so part of its expertise comes from its customers and partners. For example, deposit owners of the MagNet Bank support clients financed by the Mentor and Sector community loans. For Mentor loans, deposit owners support a specific borrower client with their savings, while in the case of Sector loans, a specific group of clients (Sector) is aided based on the nature of its activities [81]. "*We actively involve our customers into the decision-making process by our community products like Mentor/Sector deposit and loan products, or bank cards supporting social issues and helping programs, or with a donation program based on the decisions of individual customers and corporate partners. It provides for our business partners the opportunity to take responsibility, thus creating a healthier connection to money*" [72]. In November 2019, MagNet started its fourth tree planting program, and invited 150 clients to participate in the physical planting of trees. In this way, the bank provides a good example and combines its carbon-diminishing program with community building efforts [82].

The customer experience also includes other unique aspects. "*That the customer can go to a café and not to a bank branch, that we can have a meeting there, or the internet interface is very advantageous for the bank, I think.*" According to an older customer: "*the kind of alternative culture they have established in this financial sphere is, I think, slowly but surely starting to break down the walls of resistance. The idea of a café and bank branch in one, it's still a little weird though.*"

### 4.1.4. Channels

The Bank operates with a limited number of branches and offices and a rather limited number of workers. It is highly dependent on internet banking services, but as a small bank, it can have direct contact with clients, similarly to Triodos and Banca Etica [22]. They consider "*the increasing digital development of IT important, because that's the direction the world is heading, and we don't want to open branch offices either.*"

Innovative channels are key success factors in the market: "*There were things that made us market leaders, we had an award-winning mobile application and in cooperation with FINTECH we are still the only ones with an open system where two FINTECH companies are openly connected to the entire* MagNet system, and we also have its GDPR as well.*"

### 4.1.5. Key Partners

Many NGOs and companies contribute to the bank's mission. "*These strategic cooperations with our valued partners and affiliates help us increase our abilities and efforts. Together, we are able to work toward our shared goals more effectively to reinforce respective strengths through the exchange of infrastructure, partnership and networks, knowledge and expertise, as well as support the implementation of demand-driven and sustainable projects*" [72].

They have partner organizations and institutions that are experts in their respective fields, such as the International Coach Federation, the Hungarian Impact Hub, the Impact Academy, and the Non-profit Information and Education Centre. The central branch is located in an open, inclusive community centre, MagNet Community House, which hosts value-creating programs with the support of the MagNet Bank. It is "*a space that simultaneously hosts initiatives, cultural and self-awareness programs, as well as corporate events and conferences to improve society, promoting alternatives to sustainability*" [78]. Many of MagNet Bank's clients organize their programs here, which further strengthens their relationship with the bank.

Members of GABV, and GABV itself, are also key partners in sharing innovation ideas and values similar to Triodos and Banca Etica [22]. GABV provides performance measurement instruments, such as a scorecard with the most relevant dimensions of values-based banking, or the mentioned Global GHG Accounting and Reporting Standard tool and summarizing research results for assessing and enabling the development of their activities.

### 4.1.6. Key Resources

The main resources are the banking services license and banking portfolio. The shares of the bank are owned by seven equity partners, senior partners, partners (managers as small shareholders), and employees. They organize their shareholding structure to uphold the bank's mission and identity. This behavior is also observed in Triodos and Banca Etica [22]. From the very beginning, MagNet found a stock exchange listing undesirable due to the emphasis on the short-term interests of shareholders. *"The owners of MagNet Bank, whom induced the shift to ethical banking 10 years ago are still engaged personally and involved actively on operative level at MagNet Bank which ensures that the management are integrating the values-based principles in the culture and practices to all levels of organization"* [72].

Human resources also have a significant role [83] and both organizational structure (e.g., Partnership Program with employees who obtain shares and inclusion in the decision-making process) and organizational culture (e.g., open communication) are important parts of value creation.

### 4.1.7. Key Activities

As a bank, MagNet's main activity is providing financial services (credit transactions, deposit and savings accounts, and everyday payment services) in which they are a pioneer in using information technology (Netbank, Mobilebank). There are several managers who think having an internal IT development system is a "huge advantage". In its lending process, the bank uses the MagNet Impact Scorecard to perform both negative screening, in which negative social, cultural, ecological, and ethical criteria apply, and positive screening, in which the same criteria apply, to either support projects with positive impacts on sustainability and general welfare, or rule out those with a negative impact [21]. The bank makes community building a priority through its Community Donation Program, Supportive Bankcard, and Mentor/Sector loans, claiming that their *"goals are societal engagement based on transparency and education in order to raise financial awareness"* [81].

### 4.1.8. Cost and Revenues

MagNet rejects profit maximization as an end in itself. It aims for a 5–7% return, but competes with mainstream banks that achieve double-digit returns by using derivatives, credit cards, or risky financial innovations that are developed with the purpose of earning a higher financial return [24]. The approach that real economy impact is more important than money eliminates riskier, highly profitable, financial business opportunities with higher margins. Even though MagNet is a small bank, they must meet the same legal and financial requirements as their competitors. This means higher costs and lower revenues, but their performance is less volatile, which means more predictable and stable results. For example, despite the bank's efforts to implement IT improvements, it also uses suppliers, but *"delivery is increasingly slow and poor in quality . . . IT costs are rising significantly, and this is a serious problem."*

Table 2 shows an increase in the total assets of MagNet Bank from 2014 to 2018, from USD 404 million to USD 541 million. During this period, an increase of 19.4% is observed in loan to asset ratio, suggesting the firm can expect a 19.4% growth in net interest income. This increase can, however, give rise to a high level of liquidity risk. The average tier 1 capital ratio of the bank over the period was 12.92%, which supports the fact that the bank is well-capitalised, because the value is double that of the 6% minimum requirement, determined by the Basel III Accord. The return on assets (ROA) decreased from 1.48% in 2016 to 1.16% in 2018; however, the average ROA over the period increased by 0.92%. The total change in return on equity (ROE) during the studied period was 10.12%, but a decrease of 6.03% is observed from 2016–2018. This drop in ROA and ROE between 2016 and 2018 could have resulted from its 10.42% increase in loan growth. As we previously concluded, all GABV members performed well during the crisis and thereafter. *"There are early signs that investors are beginning to seek a more stable return from their investments in banks where they can also verify that their capital is being used to support real economy activity."* [56] (p. 4).

**Table 2.** Key figure for MagNet Hungarian Community Bank, 2014–2018. figures in USD million, ** based on local currency (HUF) Source: GABV (2019): http://www.gabv.org/members/magnet-hungarian-community-bank#key-figures (accessed on 12 March 2020).

| | 2018 | 2017 | 2016 | 2015 | 2014 |
|---|---|---|---|---|---|
| **Total Assets** | 541 | 512 | 421 | 380 | 404 |
| **Funds Under Management** | 0 | 0 | 0 | 0 | 0 |
| **Total Assets and Funds Under Management** | 541 | 512 | 421 | 380 | 404 |
| **Total Assets and FUM Growth (One Year) \*\*** | 14.00% | 7.49% | 12.50% | 4.65% | −10.87% |
| **Loans (net)** | 319 | 258 | 206 | 170 | 160 |
| **Loans to Total Assets** | 59.10% | 50.40% | 48.90% | 44.70% | 39.70% |
| **Loan Growth (One Year) \*\*** | 33.52% | 10.82% | 23.10% | 17.89% | −3.46% |
| **Client Funding** | 399 | 372 | 296 | 263 | 288 |
| **Client Funding to Total Assets** | 73.80% | 72.60% | 70.30% | 69.20% | 71.50% |
| **Client Funding Growth (One Year) \*\*** | 15.82% | 11.07% | 14.26% | 1.37% | 0.39% |
| **Equity** | 46 | 45 | 32 | 26 | 27 |
| **Equity to Total Assets** | 8.40% | 8.70% | 7.50% | 6.80% | 6.80% |
| **Tier 1 Capital Ratio** | 12.80% | 13.90% | 13.10% | 12.80% | 12.00% |
| **Total Revenue** | 33.5 | 31.6 | 34.6 | 22.3 | 36.9 |
| **Net Income** | 6.5 | 6.5 | 5.9 | 1.6 | 1 |
| **Return on Assets** | 1.16% | 1.35% | 1.48% | 0.46% | 0.24% |
| **Return on Equity** | 13.72% | 15.53% | 19.75% | 6.82% | 3.60% |
| **Cost to Income Ratio** | 67.10% | 106.60% | 77.10% | 87.80% | 81.20% |
| **Coworkers** | 238 | 225 | 226 | 225 | 218 |
| **Clients** | 47,750 | 42,910 | 39,051 | 34,322 | 31,838 |

*4.2. Principles of Values-Based Banking in the Operation of the MagNet Hungarian Community Bank*

The GABV principles are partly and in a complex way linked to the operation of MagNet Bank, as it has programs that represent several principles at the same time.

4.2.1. Triple Bottom Line Approach

MagNet Bank applies responsible loan and investment rules such as positive and negative filters, based on the B corporation system [84] and the MagNet Impact Scorecard. The bank only finances loans, projects, and customer needs that encourage sustainable development, environmental protection (projects that contribute to green energy and carbon emissions reduction), good general living conditions, and the protection, expansion and sustainability of natural and created assets. Thus, they consciously seek contact with companies whose loan purpose can be considered socially beneficial. One manager stresses that the scoring system "*doesn't measure the social benefits of our cute civic activity, but what actual impact our core banking activity has on the world.*"

When asked how the Triple Bottom Line approach manifests at MagNet, one of the managers highlighted the priority areas of the bank: "*What was primary, and in fact still decisive, is civic support, that is, support for civic initiatives, civic movements. The financing of alternative energies comes second, and in third place is social sensitivity. So, for MagNet Bank as a GABV member, these are the three characteristic features and basically in that order.*" Another manager emphasised the positive and negative filters used in the operation: "*We use positive-negative filters, let it be loans or account management. So, what this means is that if someone operates a tobacco factory or something that doesn't necessarily coincide with the bank's community goals, it's more than likely that we won't be able to finance it, and it is also conceivable that even their account management will be more* difficult."

4.2.2. Real Economy

Being grounded in communities serving the real economy is intrinsic for MagNet Bank. There are models in which this entails microfinancing; however, MagNet does not deal with microfinancing at all. The community nature of the bank and the representation of social values are realised within the framework of traditional banking activities, which

means that it offers a full range of traditional banking services as other commercial banks do, but along with a completely different set of values [85].

The focus of the real economy, according to one manager, is manifested in a tangible impact: "*A lot of financial institutions make a big part of their profits through different derivatives, different financial constructions that are often opaque and even lead to crises. In comparison, GAVB members specifically want to fund things that have a visible, tangible social impact. At MagNet Bank, I think that is the case with community deposits and the mentor/sphere loans and deposits.*" According to a branch manager, it is very important that "*there is no insurance at MagNet, as it would be unethical to persuade a customer to take out a new life insurance, for example, and there are only capital guaranteed products. There are only products that will not incur the customer any loss, thus there is no credit card either.*" To which another executive added: "*MagNet Bank does not offer commodity loans either, . . . the bank's management has decided that these are not compatible with the community profile.*"

### 4.2.3. Client Centred

During the interviews, the community approach was clearly evident. One of the managers comments: "*There is a so-called Community Development CEO on the Board of Directors of the Bank, which is unique in itself compared to other companies or banks, he is the one who is responsible for the internal community obviously, but also for the entire external community.*"

The bank envisions a long-term relationship with its customers. This is reflected in the personal relationship between the client and the bank clerk. While in a large bank the customer encounters different clerks, at MagNet, the client always receives service from the same clerk. As one branch manager puts it: "*While they are only a customer anywhere else, here they are THE customer.*" The bank feels a responsibility towards the customer. If a customer has trouble repaying a loan, they will try to solve the problem by rescheduling or extending the term. The atmosphere is informal and convivial, as clerks and managers often converse with clients, "*most of our clients don't choose us because of some advertising, ... but because they are referred to us.*"

The customers interviewed all agree that they maintain a very strong and personal relationship with the Bank. The reason being, according to a client, that they "*cherish the same values as MagNet, that is environmental protection, sustainable growth and so on. We are not interested in bad things that are socially useless or harmful.*" According to another customer, MagNet is "*different because it's a community bank and responsibility and sustainability are important. The most important thing for me is that there is a personal relationship between the bank and its customers. They contact me regularly, we talk about new opportunities, our relationship is very good.*" In addition to personal contact, especially for the civil sphere, it is important that "*they organize a lot of programs, which fit into our value system as well, many of us participate in them, so there is a close connection along the values.*"

### 4.2.4. Long Term Resiliency

The long-term resiliency of MagNet Bank is the result of its own conscious decisions. According to one of the managers involved in this decision "*one of the reasons for becoming a community bank was to be insensitive to the crisis.*" In 2009, Spanish and Dutch examples were studied to see how the community mindset works. It is important that there is no foreign investment, that the bank does not go public, so that even if there is a change, it does not affect the operation of the bank. According to the plan, activities include the collection of deposits and lending against collateral, then the introduction of a Community product first.

A guarantee of long-term resiliency is that the bank does not invest in risky transactions. One manager said: "*Our bank does not take part in financial speculations, our loan strategy is extremely conservative, so it does not make financial manoeuvres at all.*" In addition, the owners of MagNet Bank have clearly declared their principles of operation. This was confirmed by one of the managers during the interview: "*I think it all depends on whether the owners are satisfied with a smaller profit or not.*" Instead of focusing solely on economic profit maximization aspects, the social and environmental profit-generating effect and

ability are given the same priority. Profit expectations for a high return on equity (ROE) are significantly reduced by the owners of MagNet Bank, the desired upper level of which is set at 7%. There is a strict dividend policy: the owners will not withdraw more than 30% of the Bank's profit for a given year, 60% is spent on the development of the Community Banking model, and 10% of the profit goes to non-governmental organizations at the direct disposal of the Bank's customers. Volunteering, involvement, and responsibility prevail in both the owners' approach and the management model [86].

### 4.2.5. Transparency

In connection with transparency, we can talk about the transparency of the relationship between the owner, the customers and the employees, all of which are present at MagNet Bank. "*The owners and the management are not separated, this means total transparency towards the owners.*"

As a transparent bank, MagNet Bank offers a 'pay as you like' system to its retail customers with regard to account management fees. This enables everyone to decide responsibly what constitutes a fair account maintenance fee, within a range of 0 and 1000 forints, depending on their needs. The Bank's customers are informed about the cost price of account management (which is HUF 383 per month per retail customer), thus offering a decision-making opportunity for its partner-customers to bank for a fee in line with real costs. The customer can request the fair fee in person at the bank branch, by post, via Netbank or VideoBank [87].

In 2010, MagNet was the first Hungarian bank to calculate and inform its customers of how much they contributed to the bank's total annual profit. Customers can decide which organizations participating in the NGO program should receive 10% of their share of the profits. From the hundreds of applications received, a professional jury selects the eligible organizations, ensuring professionalism and diversity. Following the decisions of 39,646 customers, 56 social organizations received HUF 45,926,080 in 2019 [88].

The bank also strives for transparency in its adverts. "*Other financial institutions only publish their bank fees and on the other side of the announcement they put a little asterisk indicating that transaction fees are an addition. We publish fees with everything included. We communicate clearly, which is one of our most important arguments in the field of community banking, this is something we pay very close attention to.*"

### 4.2.6. Culture

All of these principles are embedded in the culture of the bank, and are best reflected in the bank's culture of working in partnership with both external and insider stakeholders. "*The MagNet brand is not just an external brand but an internal one as well.*"

The whole philosophy of the bank has significant implications in both its organizational structure and culture. A Partnership Program started two years ago: in it, equity partners meet twice a year with a changing, diverse group of lower-level managers, in order to gain insights into the bank's activity from different points of view while also letting managers know their individual opinions matter.

We conclude from our research that many consider the partnership program to be a valuable new initiative, which has increased both the horizontal and vertical flow of information. Most of the participants are proud to have been invited to attend and happy to be asked for their views and opinions. However, some feel that they must take a lot of time away from work in order to attend partner meetings. Others are also offended if they are not asked to join the program. Nevertheless, upper management attempts to involve lower level employees in the decision making process: "*I also like that in such a small bank as ours, the senior executives are there with us and they know us. . . . If there is a decision to be made, . . . we can comment on it, and they'll change it . . . They expect even from the lowest level clerks to throw in ideas, and they appreciate it.*"

The community approach is not only reflected in formal programs within the bank. A majority of employees claim that their job is good-natured and 'loose', and that it is very

far from the tight, rule-filled world of traditional banks. According to a manager who has worked at the bank for several decades "*there is also a personal bond, so the relationship and personal connections are quite close, banks usually represent the exact opposite of this worldview . . . We had a colleague who came to us from a competitor and said that for example, when he had only been with us for two months he said, "here we have human conversations."* The culture of MagNet's work environment is rather relaxed: "*People come here from other banks, new employees, and the first week they come in their suits [ . . . ] some need months, some years to loosen up, for example, one of my colleagues arrived in shorts for the first time now, although he has been working here for three years!*"

## 5. Discussion

In the following we outline the potential points of connection between the two theoretical frameworks, the BMC and GABV's six principles. They cannot be perfectly matched as the six principles are interconnected, so they have a complex impact on different parts of the business model. However, we can see patterns based on their most significantly affected areas. With the projection of the models on each other, we emphasize the role of values in the business model. Figure 3 presents the main points, but some of them are worth highlighting.

As a fundamental part of the bank's activity, the Triple Bottom Line Approach has been found to impact Key Partners, Key Activities, Value Proposition, and Customer Segments. MagNet has different priority areas than traditional banks, these are the support of the local economy (especially SMEs and NGOs) and the support for sustainability, for alternative, low-carbon energy investments, and businesses. This is the heart of value creation, and the Value Proposition, which is also connected to Real Economy and Culture principles.

We find that MagNet Bank is similar to other banks in that it creates value for Customers. However, it is unique with community products in the Hungarian financial market, though it cannot sustain itself with the latter. As a niche player, MagNet Bank serves sustainability conscious individuals, and entrepreneurs, projects, or NGOs that face difficulties in obtaining loans through the traditional banking system. As a consequence of their values and engagement, they could decide to increase the ratio of green energy and decarbonization projects within corporate loan portfolios. That is why Customer Segments can be connected to the Real Economy principle as well as the Triple Bottom Line Approach. However, as one manager put it, "*MagNet Bank's clientele does not just consist of such super-conscious people either*", so a key implication for the bank is to raise awareness, in which marketing can play a big role.

With regard to Channels and Customer Relationships, the focus should not be on account development, but on complex and sincere messages and digital solutions, in which they have had a competitive advantage for a long time. However, competition is still very intense, and it is difficult to keep up with competitors. The bank is based in Budapest, with a small presence in the countryside, which may be an obstacle to long-term development. Person-centred customer management and customer experience delivery is one of the key value creators. Online communication and information dissemination, even social education, are strongly related to transparency. MagNet Bank also supports higher education, but it would be worthwhile to move towards a strategic partnership so that the members of the next generation become responsible bank customers. Transparency also applies to owners and employees, but the main focus of MagNet Hungary is on transparency for customers who, based on the information provided, know what is happening in the bank with their money. The bank also promotes honest advertising and places particular emphasis on its support of NGOs, which is included in all of its programs. Based on its mission, the bank develops products that make customers responsible, raises their awareness, and thus contributes to solving global problems with local thinking and tools. Based on these programs and connections, clients feel that they belong to a community of responsible individuals and organizations, and can participate in decision-making. Customer Relationships and

Channels are mainly related to the Client Centred, Culture, and Transparency principles of GABV.

The key activities of MagNet are providing financial services such as credit transactions, deposit and savings accounts and everyday payment services, where they are a pioneer in using information technology (Netbank, Mobilebank). Within the traditional financial services, with services such as pay as you like accounts, the MagNet Impact Scorecard, or community programs, MagNet uses initiatives which are related to transparency and a new culture of finance. Moreover, MagNet provides community products and services, which are related to the Real Economy, but have implications on the Triple Bottom Line as well.

Its key resources are the banking services license, the banking portfolio, and employees of the banks are owned by the seven equity partners, senior partners, partners (managers as small shareholders), and employees. In recent years, MagNet has not fully complied with the regulations of the Central Bank, as it does not adopt such rigorous internal processes of regulations, as many other large banks would. However, compliance with these is essential for licensing and ensuring responsible operation, as it can minimize the risk of money laundering and other irregularities. At the same time, the commitment and common values that are embedded in a flexible and informal culture, cast doubt on the feasibility of adopting the rigid processes and habits of traditional banks. Finding a balance between the two can ensure long term resiliency.

The key partners of MagNet Bank are SMEs, NGOs, the Central Bank, and organizations which are experts in their respective fields, in addition, GABV and its members are also key partners. This relates to the Real Economy but can also be connected to the Triple Bottom Line Approach, Long Term Resiliency and Culture principles. The bank builds a true community with responsible customers and NGOs through both its products and its programs. The focus of the real economy is manifested in a tangible impact.

In terms of costs and revenues, it is essential that MagNet aims for a 5–7% return but competes with mainstream banks. MagNet is a small bank, but must still meet the same legal and financial requirements as its competitors. This means higher costs and lower revenues, but also a performance that is less volatile, which implies more predictable and stable results. The pursuit of Long Term Resiliency has been operationally important since the bank's inception. Wages are lower than the sector average, there are no foreign investments, and the bank will not go public; consequently, the owners must be satisfied with a more modest profit. The challenge lies in a cost-effective expansion, which would not only increase economies of scale and enable more competitive salaries, but also increase the number of responsible and conscious customers.

Our study provides an opportunity for values-based banks to interpret and demonstrate how GABV principles are reflected in a business model. Future research would expand the current scope and examine other values-based banks and provide a further comparison. The findings of this study can also inspire traditional banks to make their business model more sustainable. The methodology has the advantage of using the original, and well-known, Business Model Canvas concept [58], which can make it easier for businesses to apply. In addition, contrary to the triple layered business model, our methodology retains the one-page concept [18,64]. Its application is further facilitated by the fact that it allows a simple, quick overview and does not require much data, which may play a major role in establishing sustainability accounting with the development of digitalization [65,70].

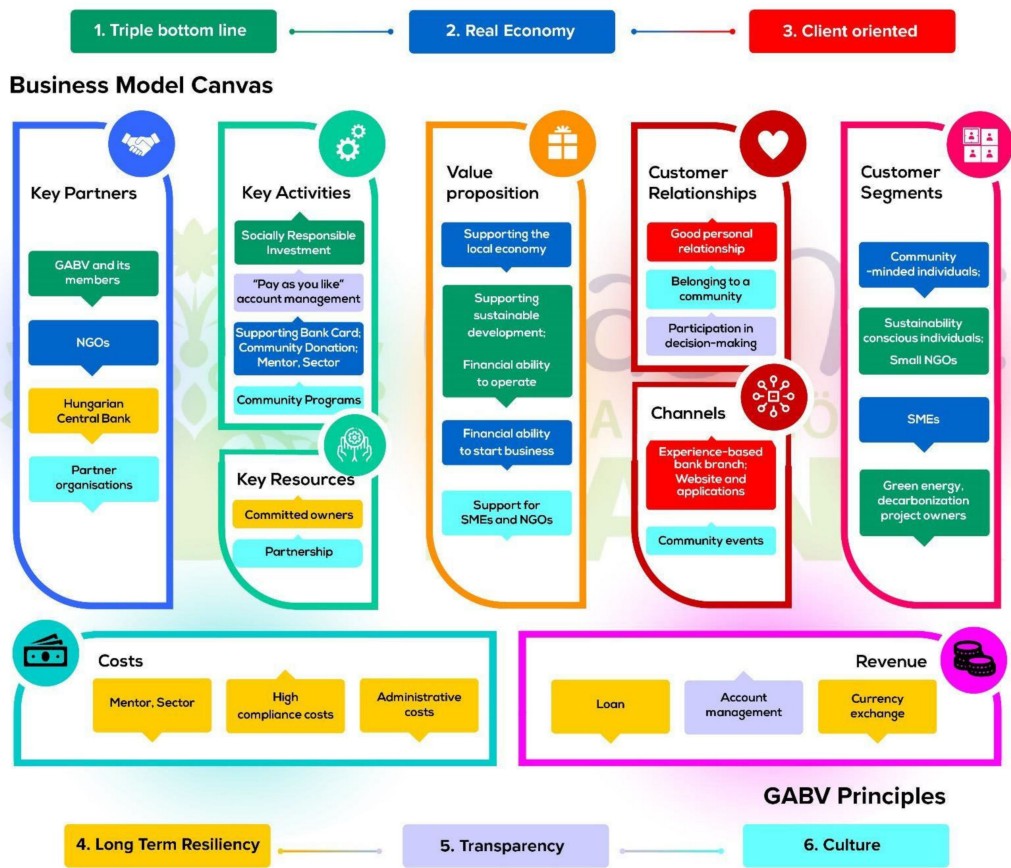

**Figure 3.** MagNet Bank's Business Model Canvas. Source: Authors' own figure.

## 6. Conclusions

MagNet Bank effectively integrates the three aspects of sustainable development into its operations, striving to take into account economic, environmental, and social aspects at the same time. Because of similarities in their missions, visions, and governing principles, MagNet and international good practice examples, such as Triodos Bank and Banca Etica, are very similar. Unsurprisingly, these three institutions are members of GABV. MagNet, however, has its own unique challenges to which it must find unique solutions. Its operations reflect the combination of the functioning of a values-based bank and a traditional bank in that it must consider the challenges of maintaining a balance between a sustainable approach and operating as a reliable and compliant bank.

We think our research, which connects the BMC with GABV principles, can help the bank to clearly plan, present, and communicate its value creation process. At the same time, the difficulties and weaknesses highlighted by the research show that MagNet Bank is struggling with the typical problems of a growing small business, which is challenged further by increasingly severe regulations in the banking sector. The question arises as to which new areas to develop, and how to obtain the necessary resources for this. The adoption of modern techniques is also a challenge for the bank. Developing mobile banking requires significant resources. Fintech techniques will reduce costs, as they eliminate the need for an extensive banking network. One of MagNet Bank's main competitive advantages is a customer-friendly service that develops personal relationships. However, this could be jeopardised by the impersonal nature of mobile banking. MagNet Bank is very similar to a social enterprise that has evolved from a non-governmental organization in that innovative community programs and culture are a kind of denial of traditional banking processes and culture. However, if this causes a detriment to important functions such as marketing, the bank will remain unable to rid itself of the stereotype of 'hippie bank' or niche market player, which implies unreliability and carries a negative connotation

in a financial sphere based on reliability and prudence. Honest and effective marketing and community building initiatives can have a very powerful impact on developing the attitudes of customers and even employees. Currently, the well-targeted marketing option is not used sufficiently by the bank. Existing customers know and appreciate the bank because of its client focused approach and sustainability-related values, but it is not enough to educate the bank's own community alone. Raising the awareness of potential customers about the value creation of the banking system and MagNet Bank as well as the Value Proposition itself could be a potential source of customer acquisition. The principles of GABV provide a compass for everyday operations and could be a valuable input for framing such a process as well. The present crisis makes consumers cautious and hopefully open to prudent, values-based business and banking. It is important to define, in the business strategy, exactly how many and which types of customers the bank wants and what steps it must take to acquire such clientele. An interesting question is how the bank envisions its longer-term future. If it succeeds in achieving its goal of making all of its customers responsible and sustainable ones, then there is a danger that MagNet's approach will be adopted by large banks that have more efficient systems in place; thus, it would lose its competitive advantage. At the same time, the owners' big dream, their mission, would remain fulfilled. The question is whether MagNet Bank will survive this economically.

During the interviews, we asked separate questions about what MagNet Bank learned from the practice of GABV banks and vice versa. It became clear, based on their responses, that the managers and employees of MagNet Bank are proud of the system in place for measuring environmental and social impacts, and there is a great deal of interest in this among GABV banks as well. At the same time, values-based banks require much closer professional cooperation between GABV banks in order to learn from each other's practices and good examples.

The MagNet Bank case study is a good example of how GABV banks can describe their business operations with a single canvas, integrating GABV principles, of how the banking sector can integrate sustainability considerations into their operations, thereby promoting sustainable development, and on how decision makers can improve compliance standards for banks to ensure not only the safe and prudent operation of the banking system, but also sustainable development.

During and after the pandemic and the resulting social and economic crisis, the clarity of business models and value creation can be a key to success. From a Triple Bottom Line and sustainability point of view, MagNet Bank is far beyond other businesses in the definition of value, but even it still has much to learn. The continuous development of all parts of the business model, based on the leading principles, is inevitable. Our belief is that such a responsible and sustainable way of doing business can help other sectors and companies cope with the global crisis and develop their capacity to bring about a smooth transition to a low carbon and sustainable economy.

## 7. Limitations and Further Research Directions

As aforementioned, the BMC has not been used much as a framework for analysing values-based banks' practice; however, it has been used for sustainability analysis in, for example, the agribusiness sector [89], the energy sector [90] and the olive oil sector [91]. Donner and Radic's [91] study of sustainability issues in the olive sector describes how *"within their Business Canvas Models, the circular bioeconomy principles' 'closing loops' and 'cascades' are principally reflected in the enterprises' key activities (collecting and re- or upcycling organic waste), key resources (using olive waste and by-products as main resources), key partners (collaborating with partners from the olive chain but also other sectors), and new value propositions (offering multiple and diversified products and services based on waste streams)"* [91] (p. 14). In their article on low-carbon infrastructure investment Foxon et al. [92] extend the original nine building blocks of the BMC for sustainability analysis by creating additional blocks within the Value proposition and Revenue streams sections. Similarly, we have undertaken the integration of the core values of the GABV model, which express sustainability

considerations, by creating additional blocks within the original BMC. In some of these articles, as in ours, the possibility of applying triple-layered BMC was also mentioned as a promising future research direction [89–91]. There are other business modelling methods that could be used, from a sustainability point of view, especially the Ecocanvas, the circular business model or the triple layered BMC. Although the integration of the BMC and the six principles of GABV make the theoretical framework complex enough (both in terms of description and visual representation), the broadening of the framework to the environmental and social canvas are a promising future research direction.

Many bankers were interviewed from all of the relevant departments, relative to the bank's size; however, the viewpoint of customers as stakeholders is not so well established. Though we interviewed members of different customer segments, we also had to rely on the bank's internal documentation, which was partly based on customer surveys, and bankers' comments on customer relations and expectations. This deficiency can be corrected in future studies.

Examining other GABV banks using the integrated approach of the BMC and GABV principles could also be a further research direction, presenting good practices, with both theoretical and practical implications.

Other limitations consist in generalisation. The study presents one case from one single country. Having recognised this risk, the authors opt for basing the research questions on theory, documenting the process in detail, and using triangulation through applying a multiple data collection (interview, content analysis, literature review) method.

Finally, our study uncovers how stakeholders are affected by values-based banks compared to traditional banks. Although we do not specifically investigate how stakeholders may be disadvantaged in either of the models, there is certainly scope for further research in this area, especially in emerging areas of greenwashing, greenscamming, and the role of regulation in values-based banks. The latter is of particular interest, given our finding of values-based banks struggling to keep a balance between regulatory compliance and a flexible and discretionary approach.

**Author Contributions:** All the authors contributed to the conceptualization, formal analysis, investigation, methodology, writing of the original draft, and writing review and editing. All authors have read and agreed to the published version of the manuscript.

**Funding:** This research was funded in the framework of the 'ISSUE—Innovative Solutions for SUstainability in Education' ERASMUS+ KA2 Strategic Partnership project (Nr 2018-1-HU01-KA202-047730).

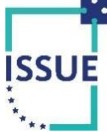

**Data Availability Statement:** Publicly available datasets were analyzed in this study. This data can be found here: http://www.gabv.org/members/magnet-hungarian-community-bank#key-figures (accessed on 11 August 2021) and https://www.gabv.org/wp-content/uploads/2016-Research-Report-final.pdf (accessed on 11 August 2021).

**Acknowledgments:** The authors acknowledge the openness and support of MagNet Hungarian Community Bank, especially Csaba Molnár, Equity Partner, Board Member, Head of Resourceful Humans and Brand Development.

**This project has been funded with support from the European Commission:** *The European Commission's support for the production of this publication does not constitute an endorsement of the contents, which reflect the views only of the authors, and the Commission cannot be held responsible for any use which may be made of the information contained therein.*

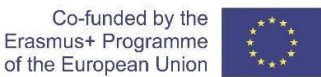

**Conflicts of Interest:** The authors declare no conflict of interest.

## Appendix A. In-Depth Interview Questions

### INTERNAL STAKEHOLDERS

1. INTRODUCTION
    1.1. How long have you been with the bank?
    1.2. Why do you work for this bank?
    1.3. Where did you work before that (especially at banks)?
    1.4. Why did you come from there?

2. BANK VALUES
    2.1. How is this bank different?
    2.2. What is different from a traditional bank?
    2.3. What is different from banking CSR in general?
    2.4. What values are important at the bank?
    2.5. What is the vision of the bank?
    2.6. What is the mission of the bank?
    2.7. What are the key strategic objectives?
    2.8. Do you want to say something about the topic of revenues/expenses that is important, but we can't find it in the annual report?
    2.9. How are the principles of GABV 6 integrated into operation?
        1. Triple Bottom Line approach
        2. Real economy
        3. Client-centred
        4. Long-term resiliency
        5. Transparent and inclusive governance
        6. Culture—How are all these principles embedded in the culture of the bank?
    2.10. Were there practices that were "learned" from other GABV members?
    2.11. Was there anything that other GABV banks might have taken over from them?

3. STAKEHOLDERS
    3.1. Who are the main stakeholders?
    3.2. What are the values that are important to them?
    3.3. What is their contribution?
    3.4. How are their needs and interests assessed?
    3.5. How do they get feedback from them?
    3.6. What are the main activities for them? (processes)
    3.7. What conflicts/problems/dilemmas did you experience in the bank?
    3.8. How were these problems solved?
    3.9. What are the most important resources?
    3.10. What are the most important activities/processes?
    3.11. Who are the most important external partners?

4. PARTNER GROUP
    4.1. How does the partner group work?
    4.2. Who are its members?
    4.3. How often do they meet?
    4.4. Can an ad hoc meeting be initiated for members?
    4.5. What questions do you usually have?
    4.6. How are issues/problems resolved?

5. MENTOR/SPHERE
    5.1. Were there any issues in the group specifically related to Mentor/Sphere deposits?
    5.2. How are issues/problems resolved?
    5.3. How does a Mentor/Sphere lending process work?
    5.4. On what basis are loan applications evaluated?

---

**EXTERNAL STAKEHOLDERS**

---

1. INTRODUCTION

    1.1.  How long has the bank been a customer of this bank?
    1.2.  Why did you choose this bank?
    1.3.  Where did you bank before?
    1.4.  Why did you come from there?

2. BANK VALUES

    2.1.  How is this bank different?
    2.2.  What is different from a traditional bank?
    2.3.  What is different from banking CSR in general?
    2.4.  What values are important at this bank?

3. STAKEHOLDERS

    3.1.  What are the most important values you expect from a bank?
    3.2.  What are your expectations towards the bank?
    3.3.  What is your contribution to the bank's value creation?
    3.4.  How are your needs and interests assessed?
    3.5.  How do you give feedback to the bank?
    3.6.  Which main activities (processes) serve your needs?
    3.7.  What conflicts/problems/dilemmas did you experience at the bank?
    3.8.  How were these solved?

4. MENTOR/SPHERE

    4.1.  Why did you choose the Mentor/Sphere deposit or loan?
    4.2.  How does a Mentor/Sphere lending process work?
    4.3.  What documents were requested from you?
    4.4.  How are issues/problems resolved?

---

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
