# Peer review of "Business Model and Principles of a Values-Based Bank—Case Study of MagNet Hungarian Community Bank"

_sustainability, doi:10.3390/su13169239_

Round 1

Reviewer 1 Report

Please revise your manuscript and present the concept based on The Triple Layered Business Model Canvas which is most suitable to the research problem you have tackled.

Author Response

Dear Reviewer,

Thank you for your review and comments concerning our manuscript entitled “Business Model and Principles of a Values-based Bank - Case Study of Mag-Net Hungarian Community Bank”. Your comment is valuable and very helpful for improving our paper. We used the suggested model as a literature reference and as a potential further research direction. For our detailed response please see the attachment.

Reviewer 2 Report

The major downside that the paper has is the overall quality of text written in English. There are numerous cases where the phrases, although written in English, lack the message. The message is sometimes unclear and confusing. I have made great efforts in some parts to read the paper and then to understand it (and I did this to grasp the idea of the research); and this was due not to the technicality of the paper, but to the bad English. One possible scenario is that this paper was first written in the native language and afterwards it was translated in English (by someone without a clear understanding on the topic of the paper). I strongly recommend that this paper must be read by an English speaking expert or native.

The authors tackle a very interesting topic that has attracted considerable debate more recently. I am somewhat sympathetic towards the basic ideas tackled in the paper. Overall, the paper is concise and relatively easy to follow. Nonetheless, although I personally enjoyed reading the paper and appreciate the topic, it’s more of a ‘light reading’. Such a manuscript usually takes the form of a chapter in a wider and more comprehensive book. It’s not an empirical work, nor a theoretical innovative one. It is, as the author him/herself mentions: “an exploratory study” (NOT an explorative study).

My comments to this paper will be made using the following annotations:

* P: page number (top right corner)

* L: line number (from the right part in each page)

Some specific comments are:

* P1, L12: its “to meet the new…”, not “for meet the new…”

* P1, L13: Instead of “Because of intermediation, the indirect, catalyst impact, banks have large responsibility and opportunity in transforming economy, e.g. by lending for decarbonization, green energy projects” I recommend “Given the process of intermediation, which as an indirect and catalytic impact, banks have a large responsibility and opportunity in transforming the economy, e.g. by lending to projects that imply decarbonization and/or green energy”.

* P1, L17: Instead of “Our conviction is that…” you can use “Our belief is that…”

* P1, L18: Instead of “as well in coping the global” you can use “as well to cope with the global”

* P1, L18: Delete “the” from “and being the part of”

* P1, L19: Instead of “This is an explorative study. Data were…” I suggest “This is an exploratory study. Data was…”

* P1, L26: “which raises need for crisis and risk management…” as “which raises the need for better crisis and risk management”

[And all these comments are only from the Abstract. There are similar mistakes/unfit expressions in the entire paper. Should I had more time, I would have performed a more thorough analysis.]

* P2, L43: there are two questions there, not one

…and so forth

Please rephrase the research questions!!!! These are very important. One suggestion is:

  1. How are the 6 Principles of GABV incorporated/embodied in the MagNet Bank operations?
  2. How can the value creation process be described using the Business Model Canvas in the case of MagNet Bank?

I end my review with three more major recommendations:

- I recommend that the authors use in the case study analysis some counterarguments, from banks that don’t accomplish the same standards as the MagNet Bank. This way the strong points the describe the values-based bank are more clearer.

- I recommend that the authors present the limitations of the paper and some ideas about the continuity.

- I recommend that the authors present in the Conclusion section the main takeaways that a banker or a policy maker (in the banking industry) can learn (using enumeration: a), b) c)…)

Author Response

Dear Reviewer,

Thank you for your review and comments concerning our manuscript entitled “Business Model and Principles of a Values-based Bank - Case Study of Mag-Net Hungarian Community Bank”. Your comment is valuable and very helpful for improving our paper. Thank you for the suggestion related to language problems. After the amendments, the article was proofread and copyedited by a native British speaker, who is a senior university researcher himself. We made all the changes based on your specific comments – thank you for the recommendations and efforts.

Reviewer 3 Report

The paper presents a case study of a Hungarian bank on its business model and principles according to values-based banks. The study is explorative in nature and interviewed bankers, costumers, and relevant documents.

The paper, in general, is a good lecture material for case studies on corporate social responsibility (CSR), sustainable banking, and sustainable finance. Hence, the topic itself fits within the scope of Sustainability journal.

Specifically,

  1. Numerous references were from gray literature. While these included the relevant bank documents for analysis, it should be balanced with academic literature.
  2. "There are few scientific articles in the literature that examine the practice of values-based banks and even fewer that analyse it using a case study method" - These "few scientific articles" should be discussed, identify what is missing, and explain how your paper fill the research gap.
  3. The research question should be placed in the Introduction.
  4. In Section 2, values-based banking and business model canvas consumed more than 25% of the main text in the manuscript. These subsection may be placed in a separate literature review section together with the relevant studies as described in #2.
  5. In addition to #4, Section 2 should focus on how the methodology was done. For example, how were the interviews done (physically, online, etc.), when, what are the interview questions, sampling technique, type/method of analysis, ethical issues, etc. 
  6. All figures should be improved. Make sure that all texts are readable.  
  7. Make sure that references are well-cited, e.g. Ching & Fauvel, 2013
  8. I'm not so sure why the authors need to specify the page number of the cited document, or whatever its purpose, e.g. L125 [36:3], L146 [9:3], L157[9:8], etc.
  9. Acronyms and abbreviations must be spelled out completely on initial appearance in text.
  10. Avoid defining acronyms many times.
  11. Avoid beginning a sentence with an acronym or an abbreviation.
  12. In L164, ROE is "return on equity" while "average annual profitability" in L172.
  13. Minor English issues in using punctuation marks, articles, etc.

Author Response

Thank you for your review and comments concerning our manuscript entitled “Business Model and Principles of a Values-based Bank - Case Study of Mag-Net Hungarian Community Bank”. Your comments are valuable and very helpful for improving our paper. For our detailed response please see the attachment.

Reviewer 4 Report

sustainability-1257061

This manuscript addresses unique and original subject matter involving the MagNet Hungarian Community Bank’s approach as a values-based bank for introducing how the operation of the bank differs from that of traditional ones. Based on interviews of 11 bankers and 3 customers, qualitative aspects of stakeholder views on green and ethical banking are identified, analyzed and discussed. Overall, the manuscript is well written and provides an important analysis of actual stakeholders  (bankers and customers) as they address contemporary issues and concerns around SARS-COV-2 and climate change.  The main concern is that the banking practices and results are not evaluated against Environmental, Social, and Corp Governance (ESG) criteria or address the Social Development Goals (SDGs). I suggest that a simple table summarize these aspects in the results section. This addition should expand reader interest. 

1) While the approach is original and informative, the number of customers is small, limiting the potential knowledge base for this stakeholder group. Please include several lines addressing the limitations of the study. 

2) A simple stakeholder analysis should be considered in this study to more clearly describe which stakeholders benefit and/or are disadvantaged relative to financial terms. The user customer cases are clear--are there anti-user cases such as greenwashing that should be considered?

3) The GABV material is redundant across sections and could be shortened.

4) Reduce the number of keywords.

5) Line 46, define CSR.

6) Shorten the materials and methods section. Section 2.1 reads as additional introductory material and should be heavily revised.  In this section, very briefly, describe only the exact documentation, interview methods, and model description employed. The model Canvas section is quite clear and short.

7) Figure 1 does not fit in Materials and Methods. It could be a “Box” graphic placed in the introduction as it describes well the subject matter of the study.

8) Table 1 is a result, not a method, and should be placed in the results section.

9) Overall, the results, discussion, and conclusion sections are well written but could be shortened.

10) Can you provide, briefly, how the results for the MagNet Hungarian Community Bank compare to other EU banks? 

Author Response

Dear Reviewer,

Thank you for your review and comments concerning our manuscript entitled “Business Model and Principles of a Values-based Bank - Case Study of Mag-Net Hungarian Community Bank”. Your comments are valuable and very helpful for improving our paper. For our detailed response please see the attachment.

Round 2

Reviewer 1 Report

In abstract the Authors declare to use Business Model Canvas methods to examine Mag-17 Net Hungarian Community Bank’s approach as a values-based bank. Values based banking is a kind of banking that focused on values created mainly by nonfinancial factors –like social and environmental. Business Model Canvas is a method that based on and analyses economic factors like  partners, resources, channels, customers etc. and as a result diagnoses cost and revenues. How the Authors want to analyse nonfinancial factors and sustainable value based on the method (BMC) design for analysis of economic / financial factors ?. The methodology is not suitable for the specific of the research problem because the method BMC is limited and omits social and environmental aspects that are crucial for value based banking.   The Authors declare “For sustainability-focused banks, profit is a result of sustaining and growing value in the real economy and healthy communities, not an end goal” MagNet Hun-329 garian Community Bank defines value as such: “Value based principles are at the heart of business model and appears of all 330 levels of the organizations, and also through products and services we are creating involvement for employees, customers, social  entrepreneurs and NGO partners. With our financial services, we work day by dayday-in day-out towards the establishment of a  transparent, value-centric society.” How the BMC analysis and measure value-centric society ? how the Authors confirm this ? again “MagNet Bank develops products that make its customers responsible. This  means developing financial awareness and contributing to a democratic, sustainable, harmonious society through widespread social awareness.” Where you can confirm this by BMC ? there are many others examples of serious flaws. I don’t understand the sentence “As we indicated, BMC has not been much used much as a framework for analysing values-based banks’ practice” in the context BMC is the leading method mentioned in the abstract ? I disagree with the statement “Although the integration of the BMC and the Six Principles of GABV makes the theoretical framework complex enough (both in terms of description and visual representation), the broadening of the framework to the environmental and social canvas as well is a promising future research direction” Why GABV ? what is the argument supporting this choice and why the Authors didn’t mention about it in the abstract, it is confusing and the paper is difficult to follow as the most important information is at the end or in the middle of the paper. The Authors didn’t provide clear arguments for the methodology and the methods they used.

Author Response

Dear Reviewers,

Thank you again for your review and comments concerning our manuscript entitled “Business Model and Principles of a Values-based Bank - Case Study of MagNet Hungarian Community Bank”.

We are glad that from most of the Reviewers our paper got a favourable second round review. The Reviewers’ suggestions significantly contributed to improving the quality of information in the paper and to offering a better understanding of the case and its further implications. It was reassuring to read that in the Reviewers’ opinion “…its (the paper’s) content is significantly better than the original submission… I am looking forward to read the published version of the paper” (Reviewer 3) and “I believe that this version of the paper is considerably improved. Overall, the paper is ready to be published”. (Reviewer 2) At the same time, your new comments are all valuable and very helpful for revising and improving our paper. We checked again and corrected the references and minor English comments. We thank for the suggestion of Reviewer 2 and Reviewer 3 regarding it.

We would like to especially thank you, Reviewer 1 for your comments, which again gave us food for thought and we hope improved the clarity of the paper.  Related to your specific comments our answers are in the attached file.

We highlighted the revisions using the Track Changes function in Microsoft Word. In addition, we have highlighted the new changes in yellow.

Thanks again to all our Reviewers for their efforts and valuable suggestions.

Reviewer 2 Report

I appreciate the effort made by the author to tackle all my suggestions. I believe that this version of the paper is considerably improved. Overall, the paper is ready to be published. In addition, I highlight the following: check all your references again! It is very important to have your references updated and correctly written.

Author Response

Dear Reviewers,

Thank you again for your review and comments concerning our manuscript entitled “Business Model and Principles of a Values-based Bank - Case Study of MagNet Hungarian Community Bank”.

We are glad that from most of the Reviewers our paper got a favourable second round review. The Reviewers’ suggestions significantly contributed to improving the quality of information in the paper and to offering a better understanding of the case and its further implications. It was reassuring to read that in the Reviewers’ opinion “…its (the paper’s) content is significantly better than the original submission… I am looking forward to read the published version of the paper” (Reviewer 3) and “I believe that this version of the paper is considerably improved. Overall, the paper is ready to be published”. (Reviewer 2) At the same time, your new comments are all valuable and very helpful for revising and improving our paper. We checked again and corrected the references and minor English comments. We thank for the suggestion of Reviewer 2 and Reviewer 3 regarding it.

We highlighted the revisions using the Track Changes function in Microsoft Word. In addition, we have highlighted the new changes in yellow.

Thanks again to all our Reviewers for their efforts and valuable suggestions.

Reviewer 3 Report

The authors addressed the comments of the reviewers.

While the paper is difficult to read due to the tracked changes format, its content is significantly better than the original submission. 

Only minor comments in English (probably this is due to the tracked changes which I find it difficult to read) as well as the section numbering must be checked before its acceptance for publication.

I am looking forward to read the published version of the paper.

Author Response

(The authors gave the same response as above.)

Round 3

Reviewer 1 Report

“Related to criticisms on the usage of Business Model Canvas (BMC): in our opinion, the BMC is a framework that describes the business model of an enterprise. The way how the enterprise runs its business can be demonstrated in the BMC by the content the enterprise (or researchers) fills the model with. Values-based banks (of which the main global association is GABV) integrate the triple bottom line approach, the three dimensions of sustainability into their way of doing business, into their everyday operations. This is the very distinctive nature of values-based banking compared to the CSR of traditional banks: while traditional banks do not change the core business, they only make a few gestures towards society and the environment, values-based banks use the entire operation: value proposition, customer relationships and the entire customer side, as well as the potential of resources, activities and part” Dear Author please provide strong evidence for your arguments based not on your opinion but the state of art and related work. Please don’t provide the definitions I know them again please provide examples of research methodology dealing with three dimension perspective (ESG) and based on BMC?

“You are right, BMC is a method that originally based on and analyses economic factors, but over the years, it has proved to be a useful tool for several researchers to integrate non-business and non-financial issues, even norms and values”. Please provide evidence the list of the papers.

“As we wrote (page 6, lines 198-199) “Business modelling is a popular way of both business planning and delving into the essence of certain ventures.”, but we found only one article which used BMC for analysing the operation of values-based banks (page 7, Scheire, C.; S De Maertelaere, S. Banking to make a difference. A preliminary research paper on the business models of the founding member banks of the Global Alliance for Banking on Values. Research paper, Artevelde University College Gent 2009, June, 2009.), that is one reason why our research is a contribution to the topic”. Your statement back up my thesis your methodology is not a proper one for the research problem you analysed

“GABV is the global alliance of values-based banks, its members and the organization itself are very important partners of MagNet Hungarian Community Bank, and the core values of MagNet are defined based on the 6 Principles of GABV as well. For clarity, we have highlighted in the abstract that GABV is the main association of values-based banks. (See lines 23-24)

Based on your comment, we integrated one sentence from Discussion (See lines 25-26) and one from Conclusion (See lines 28-29) to the Abstract.” You provided definition but still there is no explanation why GABV ? please explain

Author Response

Dear Reviewer,

We tried our best to improve the manuscript and made the required changes based on your comments and suggestions. We appreciate for your warm work earnestly and hope that the correction will meet with approval. Please find our detailed answer in the attached file.

Yours sincerely,

The Authors
